# The use of non-model *Drosophila* species to study natural variation in TOR pathway signaling

**Tessa E. Steenwinkel**, **Kailee K. Hamre**, **Thomas Werner** *

Department of Biological Sciences, Michigan Technological University, Houghton, Michigan, United States of America

* twerner@mtu.edu

## Abstract

Nutrition and growth are strongly linked, but not much is known about how nutrition leads to growth. To understand the connection between nutrition through the diet, growth, and proliferation, we need to study the phenotypes resulting from the activation and inhibition of central metabolic pathways. One of the most highly conserved metabolic pathways across eukaryotes is the Target of Rapamycin (TOR) pathway, whose primary role is to detect the availability of nutrients and to either induce or halt cellular growth. Here we used the model organism *Drosophila melanogaster* (*D. mel.*) and three non-model *Drosophila* species with different dietary needs, *Drosophila guttifera* (*D. gut.*), *Drosophila deflecta* (*D. def.*), and *Drosophila tripunctata* (*D. tri.*), to study the effects of dietary amino acid availability on fecundity and longevity. In addition, we inhibited the Target of Rapamycin (TOR) pathway, using rapamycin, to test how the inhibition interplays with the nutritional stimuli in these four fruit fly species. We hypothesized that the inhibition of the TOR pathway would reverse the phenotypes observed under conditions of overfeeding. Our results show that female fecundity increased with higher yeast availability in all four species but decreased in response to TOR inhibition. The longevity data were more varied: most species experienced an increase in median lifespan in both genders with an increase in yeast availability, while the lifespan of *D. mel.* females decreased. When exposed to the TOR inhibitor rapamycin, the life spans of most species decreased, except for *D. tri*, while we observed a major reduction in fecundity across all species. The obtained data can benefit future studies on the evolution of metabolism by showing the potential of using non-model species to track changes in metabolism. Particularly, our data show the possibility to use relatively closely related *Drosophila* species to gain insight on the evolution of TOR signaling.

## Introduction

The fruit fly has been extensively used as a model to understand cellular pathways in eukaryotic organisms, specifically *Drosophila melanogaster* (*D. mel.*), whose genome harbors orthologues of about 70 percent of all human disease-causing genes [1]. Among these genes are those

**Data Availability Statement:** All relevant data are within the paper and its Supporting Information file.

**Funding:** This research was funded by NSF grant #DOB/DEB1737877 to TW as well as the Barry

Goldwater Fellowship, a Michigan Tech SURF Award, and a Michigan Tech Songer Research Award. The funders had no role in study design, data collection and analysis, decision to publish, or preparation of the manuscript.

**Competing interests:** The authors have declared that no competing interests exist.

that make up one of the most prominent metabolic pathways found in all eukaryotic organisms, the Target of Rapamycin (TOR) pathway. The TOR pathway is a central metabolic pathway linking nutritional inputs to cellular growth and proliferation. As the TOR pathway is highly conserved among eukaryotes, including mammals (mTOR) and *Drosophila* (dTOR), *Drosophila* species provide the opportunity to study TOR pathway regulation and the possible evolution thereof without the ethical complications [2–4].

Despite the advantages of the *Drosophila* model, dietary and metabolic studies in *D. mel.* have been met with inconclusive results regarding the effect of nutrition on longevity and fecundity [5–10]. Hence, additional studies are needed to improve our understanding of all factors involved in TOR pathway signaling. The current study focuses on the direct influence of nutrition on longevity and fecundity. In addition to *D. mel.*, we studied three non-model *Drosophila* species with diverse feeding habits to assess if evolutionary changes have impacted the metabolic response to nutrition via TOR signaling. In diverse *Drosophila* species, it can be assumed that TOR signaling is mostly conserved, although some changes in its regulation can be expected due to the unique environmental niches they inhabit and the wide variety of food sources available. While natural nutritional inputs are diverse, in our study we focused on one particular type of yeast to regulate the amino acid and mineral intake of the flies, thereby activating the TOR pathway. To determine the effects of nutrition on our four fruit fly species, we measured the adult lifespan [11, 12] and the number of eggs that the females laid as a proxy for female fecundity [12].

The TOR pathway is a key metabolic pathway that regulates cell survival and proliferation in all eukaryotic cells. Despite its deep conservation, all pathways are subject to mutations and evolutionary changes, especially those that connect as many inputs as the TOR pathway does. The activation of the TOR pathway is mostly initiated through sensing nutrients in the cell's environment. The available nutrients activate TOR complex 1 (TORC1), which in turn activates TORC2. Both complexes are made up of the Tor protein itself plus additional proteins that differ in their composition among the two complexes. The activation of TORC1 results in a shift of the cell to expend energy, primarily leading to an increase in protein synthesis as well as a decrease in apoptosis [13]. The higher protein concentration and availability of nutrients allow for an increase in the formation of new gametes [14]. The subsequent activation of the TORC2 complex is directed toward cell mobility and cell division, which is characterized by an increase in the production of cytoskeletal components. In the absence of TOR signaling, cells go into maintenance and cleanup mode. In general, the inactivity of the TORC1 complex has been linked to a lifespan increase, which involved its downstream targets 4E-BP being activated and SK6 inactivated [13, 15–17].

The regulation of TOR in drosophilids is critical for the proliferation and maintenance of germline stem cells, such as in the ovaries of *D. mel.* [18]. While *Tor* null mutant flies are not viable, hypomorphic *Tor* mutants showed small and distorted ovaries [19]. TOR signaling can further be manipulated through the diet by altering the quantity of nutritional protein. Through the addition of nutrients to basic agar, Drummond-Barbosa and Spradling (2001) were able to increase the rate of egg production in *D. mel.* by 60-fold [7]. However, TOR is not the only factor regulating the proliferation of germline stem cells, as ovary follicle maintenance is independent of TOR [20]. Studies going beyond the regularly used model species may reveal important insights into the involvement of TOR signaling in reproduction and lifespan.

The TOR pathway can be blocked by feeding animals with the TOR inhibitor rapamycin. This inhibitor was discovered through its ability to pause various metabolic diseases by halting cell proliferation through the inhibition of TORC1 [21, 22]. Earlier studies performed in *D. mel.* showed that introducing rapamycin into the diet can cause a decrease in female fecundity as well as an increase in lifespan in flies of both genders [23]. The observed lifespan increase

may be a side effect of starvation, which itself causes the cells to halt growth and proliferation, thereby shifting towards cellular maintenance and cleanup [24]. However, the relationship between rapamycin and longevity has been disputed in additional studies, possibly due to the confounding effects of rapamycin and nutrient concentration. Flies on low-yeast diets experienced a lifespan extension on lower rapamycin concentrations [25], while on higher yeast quantities and a higher rapamycin concentration, a decrease in lifespan was observed [23]. Thus, comprehensive studies are needed that test the effects of nutrition on life-history traits in a more controlled manner.

In the current study, we used four species of *Drosophila* with different nutritional preferences and fed them with standardized yeast amounts to observe their responses to nutritional intake. These species include *D. mel.*, *Drosophila gutiffera (D. gut.)*, *Drosophila deflecta (D. def.)*, and *Drosophila tripunctata (D. tri.)*, which belong to the Sophophora subgenus (*D. mel.*) and the immigrans-tripunctata radiation (*D. gut.*, *D. def., and D. tri.*) [26], respectively. *D. mel.* is most frequently found around decaying fruits and vegetables [27] and has been characterized as an omnivore [28]. *D. gut.* is a member of the quinaria species group and uses mushrooms as its primary food source. It prefers laying its eggs on mushrooms when given the choice of mushrooms, bananas, tomatoes, lettuce, or plain agar [29]. It is likely to breed in toxic mushrooms and is tolerant of the mushroom toxin alpha-amanitin [30]. *D. def.*, although also being a quinaria group member, does not breed in mushrooms but prefers rotting plant leaves instead as a breeding site, especially those of yellow water lilies [31, 32]. The fourth species in our study is *D. tri.*, which belongs to the tripunctata species group. It can utilize both decaying fruit and mushrooms, including some toxic mushrooms, putting its dietary preferences between those of *D. mel.* and *D. gut.* [33, 34].

Here we show that four ecologically diverse *Drosophila* species display slight variations in responses to TOR activation through a standardized nutritional yeast source and to TOR inhibition by rapamycin. These differences may suggest changes in TOR pathway regulation between the species as a result of adaptation to different lifestyles since they diverged from a common ancestor approximately 47 million years ago [35].

## Materials and methods

### Fly stocks

All fly stocks were maintained at room temperature in 1-pint bottles on cornmeal-sucrose-yeast medium [31] at a 12:12 h day/night cycle. The following three wild-type fly stocks were a kind gift from the Sean B. Carroll lab at UW-Madison: *D. mel.* Canton-S, *D. gut.* #15130–1971.10, and *D. def.* #15130–2018.00. The *D. tri.* wild-type stock E14#3 was collected by us in Escanaba (Michigan), and later donated to the National *Drosophila* Species Stock Center.

### Fly collection for dietary response experiments

In order to obtain freshly hatched flies, parental flies were allowed to lay eggs for three days in 1-pint bottles at room temperature on a 12:12 h day/night cycle, thus avoiding overcrowding in the offspring. Newly eclosed flies from these cultures were collected in the late afternoon and incubated overnight in 9-cm plastic Petri dishes (Fisher Scientific) containing plain sugar agar medium [31] to eliminate the flies that die shortly after eclosion. On the following morning, ten healthy flies of each sex were combined and placed into 3.5-cm plastic Petri dishes (Fisher Scientific) containing sugar agar with different combinations of rapamycin and yeast powder. All experiments were replicated thrice.

## Yeast quantity measurements for dietary response experiments

Fleischmann's active dry yeast was ground to a fine powder with a mortar and pestle and stored at 4˚C until further use. In order to feed our flies with different yeast quantities, we measured the yeast powder using micro curettes (Fine Science Tools). The smallest available micro cuvette featured a tip diameter of 0.5 mm (#10080–05), which equaled a volume of 250 nL, hereafter referred to as one scoop. Larger micro curettes were used to measure larger yeast volumes: model #10081–10 held 5 scoops at a time, while 6 servings using model #10083–20 amounted to what we refer to as *ad libitum* yeast on a single 3.5-cm plate.

For the experiment using heat-killed Baker's yeast, we baked the instant yeast grains at 200˚C for 30 min. We tested the yeast by suspending some baked yeast in water and incubated it for 60 min at 37˚C, with active yeast grains used as a control. The absence of gas bubbles in the baked yeast suspension was used as an indicator that the yeast had been fully inactivated.

## Rapamycin addition to plates for dietary response experiments

Rapamycin powder (>99%, #R1017-200mg, Fisher Scientific) was dissolved in the ethanol/ tegosept solution [31] that is usually added to the cooling sugar agar medium before the plates are poured. We set the final rapamycin concentration to 200 μM in the agar, based on studies showing that this concentration yields the bioavailability in flies, which is also commonly used to study TOR signaling in mice [6, 36].

## Experimental procedure for the dietary response measurements

We used two sugar agar medium conditions on our plates, which were either without rapamycin or with 200 μM of rapamycin. The yeast quantities ranged from no yeast, followed by incremental increases measured in [scoops per still living fly], to *ad libitum* yeast, where the flies had more yeast than they could eat within 24 h. Fresh plates were always prepared a day in advance, refrigerated, and brought back to room temperature one hour before transferring the flies on the next day.

On Day 0, 10 males and 10 females were combined on each plate. On each day thereafter, until the last fly had died, the flies were moved to fresh plates (of the same dietary condition), using $CO_2$ gas. Each newly vacated plate from the previous 24 hours of incubation was inspected for eggs. To collect fecundity data, the eggs on each vacated plate were counted after arranging them into 10x10 rows and columns. To measure longevity, we daily recorded the number of dead flies and their sex on each plate.

## Ovary dissection and imaging

Under the general conditions described in the previous sections, 10 male and 10 female flies were reared on four different diets: 1) 2 scoops of yeast per fly without rapamycin, 2) 2 scoops of yeast per fly with rapamycin, 3) *ad libitum* yeast without rapamycin, 4) *ad libitum* yeast with rapamycin. The flies were transferred daily to fresh food plates of the same kind for 5 consecutive days, after which female flies were sacrificed and their ovaries dissected in 1X PBT under a dissection microscope in a 9-well plate. All images were taken with a dark-field filter at 25X magnification.

## Statistical analyses

Statistical analyses were performed in Microsoft Excel and R Software, primarily using base R with the stats and survival package. The Excel stats package was used to calculate mean, standard error of the mean (SEM), median, skewness, and standard deviation values for longevity

experiments. The R stats package was used to perform ANOVA tests with post hoc TukeyHSD. TukeyHSD of multiple comparisons of means with 95% family-wise confidence levels resulted in significance codes of $p<0.001$: '***', $p<0.01$: '**', $p<0.05$: '*', $p<0.1$: '+'. To normalize the measure of egg yields across the species, the egg yields of the first 10 days of egg production were tracked. The 10-day period started when all three replicates produced eggs simultaneously. For longevity calculations, 30 females and 30 males were monitored in three replicate plates, each holding 10 female and 10 male flies. Yeast quantities were adjusted daily based on the number of living flies still remaining from the original 20. Data for the median age combined all three replicates for an overall n = 30 females and n = 30 males. Any accidental deaths were disregarded from the longevity data and adjusted for in the mean egg production averages. Results were analyzed using the R survival package, allowing for the graphing of survival curves and the calculation of pairwise comparisons, using the Log-Rank test with p-value adjustment with BH.

## Results and discussion

### Validation of the effect of yeast on the fecundity and longevity of *D. mel*

It is known that *D. mel.* does not produce eggs in the absence of dietary yeast, and that female fecundity and longevity are positively correlated over a moderate yeast quantity range, until female longevity decreases at high yeast quantities [10, 37]. Our first goal was to reproduce these findings from the literature, using our *D. mel.* strain Canton-S. We treated *D. mel.* adult flies of both sexes with five quantities of live Baker's yeast diets, including no yeast, 1/4th of a scoop (62.5 nL) per fly, 1/2 of a scoop (125 nL) per fly, 1 scoop (250 nL) per fly, and an *ad libitum* amount. Our results corroborated the published data: females without yeast did not produce eggs, while the egg production was positively correlated with the amount of yeast available to the flies, peaking at the *ad libitum* amount with an average of 37 eggs per female per day (Fig 1B). For our fecundity analysis of the same experiment, we focused primarily on the first 10 days of reproductive activity, which are characterized by a peak in egg production, followed by a plateau and subsequent decline in egg-lay activity as the flies age (Fig 1A). We also saw that females on plates with any amount of yeast started producing eggs within the first day.

The longevity aspect of our experiment also confirmed the data in the literature. While increasing amounts of yeast positively correlated with the mean longevity (in both sexes), female longevity significantly dropped at the *ad libitum* quantity (Fig 1C, S1 Fig). At this high amount of yeast, the egg production was approximately four times higher than on the second-highest amount of yeast (1 scoop per fly per day) (Fig 1B). Several other studies performed in *D. mel.* also confirmed that female longevity decreases when laying large numbers of eggs [10, 38, 39]. We speculate that the observed female-specific drop in longevity on high-yeast diets is due to the stress caused by the excessive egg production as well as mating frequency [37]. Thus, there is a trade-off between longevity and fecundity: *D. mel.* females on a rich yeast diet produce many eggs, but their lifespan is reduced.

### Effect of yeast quantities on fecundity and longevity of non-model *Drosophila* species

Our next question was if different (non-model) *Drosophila* species behave in the same way as *D. mel.* to nutritional stimuli or if we can identify differences, which may be due to evolutionary changes in TOR signaling or other nutrition-sensing pathways among these species. We know that there are high levels of TOR pathway conservation among eukaryotes; it has even

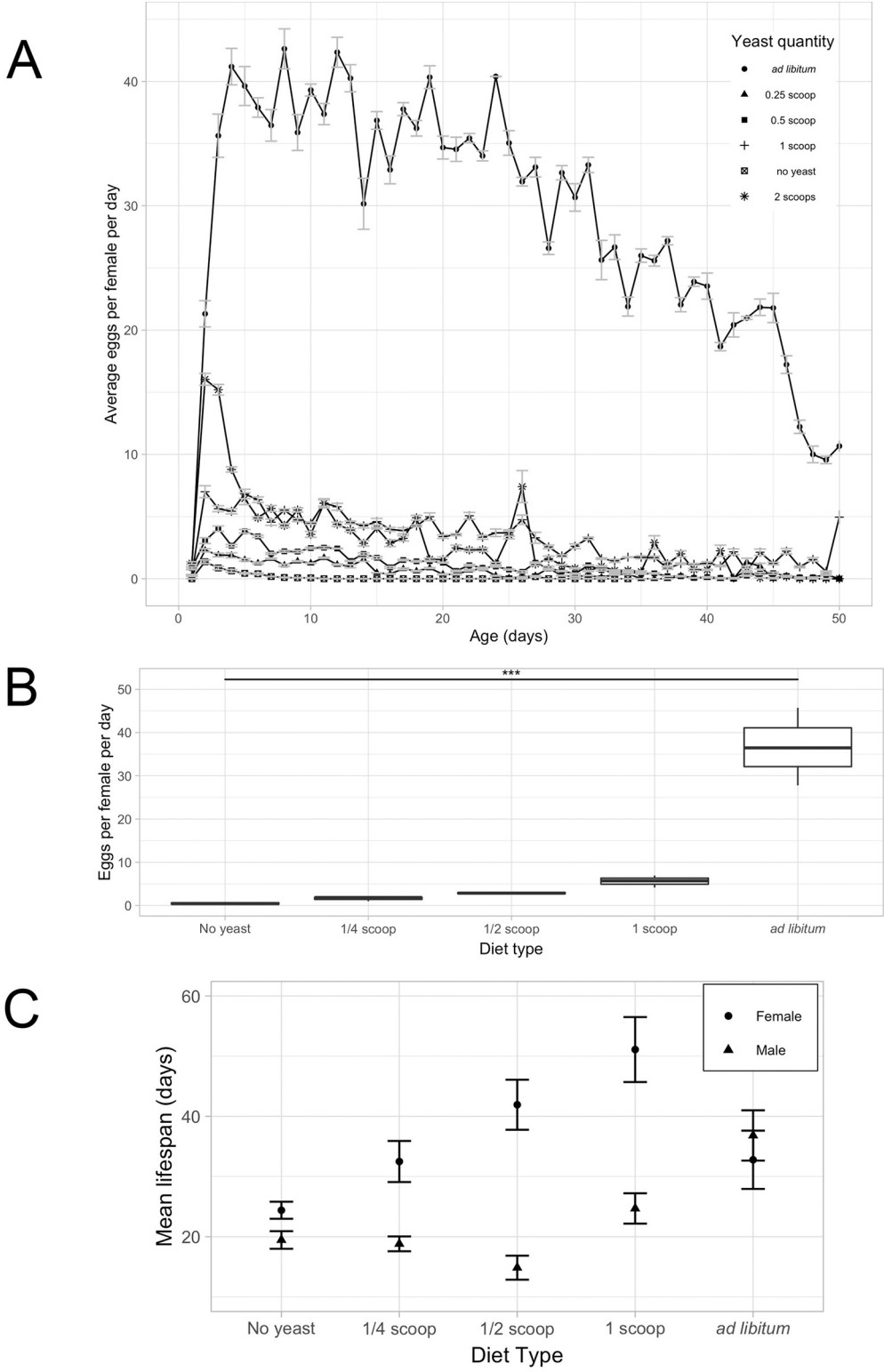

**Fig 1. Dietary effect on the average fecundity and longevity of *D. mel*. A)** Average number of eggs laid per female per day across the first 50 days of adult lifespan on five dietary treatments: no yeast, ¼ scoop of yeast (62.5 nL) per fly per day, ½ scoop per fly per day, 1 scoop per fly per day, and *ad libitum* yeast. Error bars indicate the SEM **B)** Average number of eggs laid in the first 10 reproductive days per female of *D. mel*. on diets described in A. p<0.001: '***'. **C)** Mean female (circles) and male (triangles) lifespan on the diets described in A. Error bars indicate the SEM.

been hypothesized that the TOR pathway pre-dates the insulin signaling pathway due to the absence of insulin signaling in yeast and *Arabidopsis*. However, there are regulatory differences between dTOR and the TOR systems found in *Caenorhabditis elegans* and *Arabidopsis* [40], which led us to ask how TOR regulation might be different among closely related species. We chose three additional species that can be easily reared on yeast in the lab, although they have quite different food preferences in nature: *D. gut.*, *D. def.*, and *D. tri.*, while *D. mel.* served as a control. The dietary treatments in this experiment ranged from no yeast, minimum (1 scoop/ fly/day) yeast, to an *ad libitum* yeast quantity. As a result, we observed that all four species completely ate the minimum amount of yeast in each 24-hour cycle. On the *ad libitum* yeast diet, there was of course much yeast left after 24 hours, but the amount of feces left on the plates indicated that all species ate comparable amounts of yeast. Our three non-model species differed most prominently in the maximum rate of egg production on the *ad libitum* diet, while the general trend remained the same: no yeast resulted in no eggs, little yeast in few eggs, and the *ad libitum* yeast diet allowed the maximum production of eggs in each species, which was in each case much higher than on the minimum diet (Fig 2). While all four species of flies have different lifestyles and possibly react differently to mating levels depending on yeast availability, yeast availability and the generation of offspring seem tightly correlated. The data are shown as the average number of eggs per female per day over the first 10 reproductive days. *D. mel.* laid the largest number of eggs (40), which confirms the number obtained in our first experiment quite well (37) and the results of additional studies [7, 23]. With an average of 17 eggs, the egg-lay capacity of *D. gut.* and *D. def.* was only about half as high as that of *D. mel.*, while *D. tri.* showed the lowest fecundity on the *ad libitum* yeast of about 7 eggs. In summary, our data show that all species follow a similar trend of an increased egg production with an increase of yeast availability, while the extent to which egg-laying is increased differs quantitatively by species, *D. mel.* being the most egg-productive species. This interspecific difference in fecundity could hint at a possible difference in the regulation of the TOR pathway, or in how sensed nutrition leads to the production of offspring [41].

From the same experiment, we also collected longevity data simultaneously (Tables 1 and 2, S2 Fig). In accordance with the results of our first experiment with *D. mel.* only (Fig 1), the longevity of our control *D. mel.* female flies increased with small amounts of available yeast and again decreased on the *ad libitum* diet, with many females dying in the first 21 days (Table 1, S1 Fig). In all other species, we detected an increase in lifespan as the availability of yeast increased, but most importantly, we no longer observed a decrease in female life expectancy as the diet increased to an *ad libitum* quantity. Instead, all females showed significantly longer lifespans on the *ad libitum* diet, as compared to the same species on minimum food (S1 File). As for the males, all species showed a significant continuous increase in lifespan as yeast concentrations increased (Table 2, S2 Fig, S1 File). The increased lifespans show the close connection between nutrition and fecundity, likely mediated through TOR. Overall, females usually outlived their conspecific males, except for *D. def.* Besides the observed trends, the mean lifespans of the flies across diets also differed significantly. While *D. mel.* females only lived a maximum of 63 days on an *ad libitum* diet, some *D. gut.* females lived for over 125 days on the same diet. For males, we observed the shortest mean lifespans of 10 days for *D. gut.* on no yeast, while the longest-living males were of *D. def.*, which lived for up to 98 days on an *ad libitum* diet. The longevity data among species are of importance as they allow for the examination of trends previously only seen in *D. mel.* Most uniquely, the reduction in lifespan on an *ad libitum* diet is only evident for females of *D. mel.* This lifespan reduction has been discussed as being a fitness cost [42] and a cost of reproduction [43], the stress factors being the egg production itself as well as the exposure to males. An additional factor in reducing longevity could be an increased mating frequency due to increased yeast, in turn leading to increased offspring.

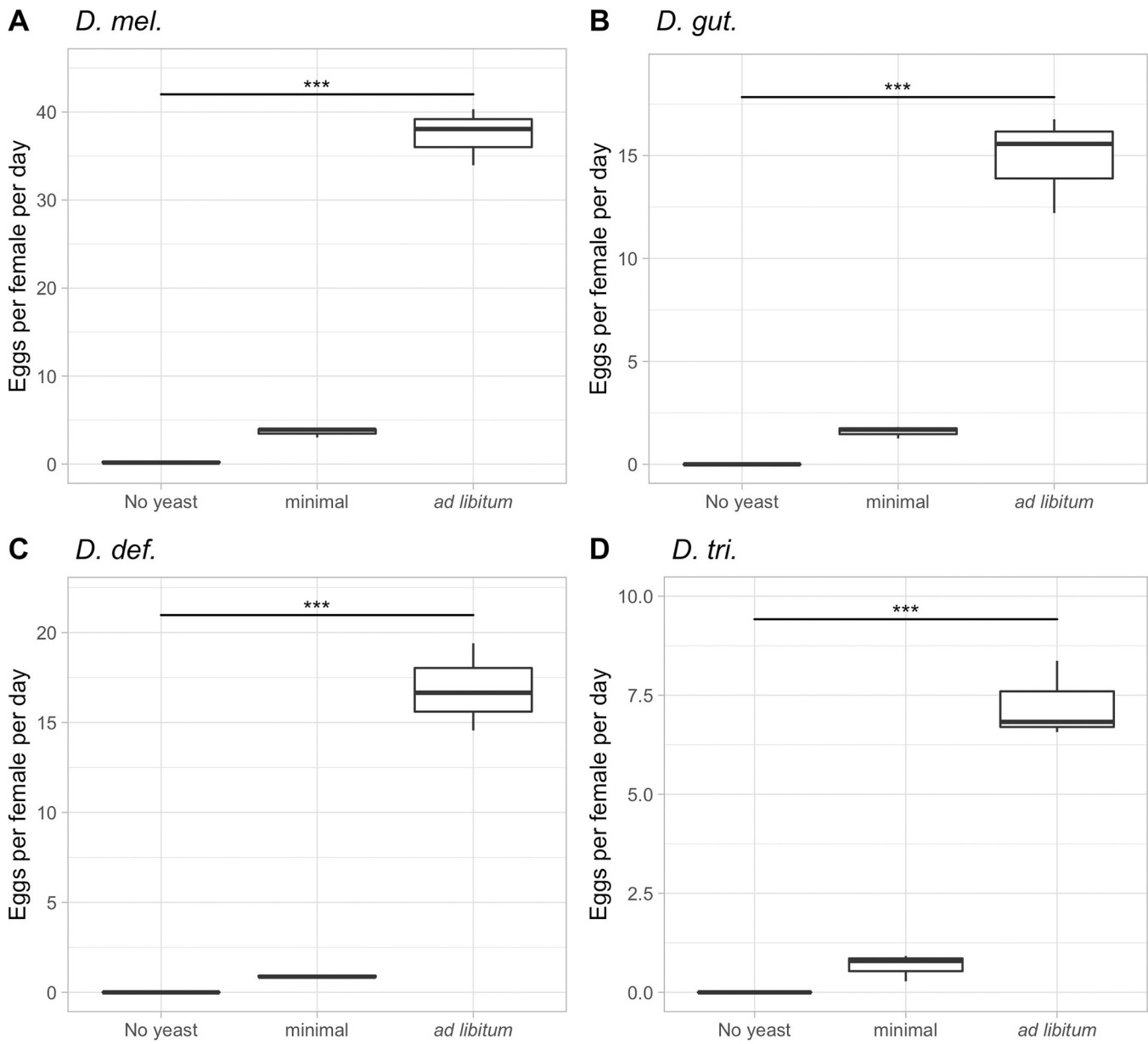

**Fig 2. Dietary effect on the average egg-lay rates of four *Drosophila* species. A)** Average number of eggs laid in the first ten days per female of *D. mel.* on three dietary treatments: no yeast, minimum yeast (1 scoop/fly/day), and *ad libitum* yeast. **B)** identical to A but using *D. gut.* **C)** *D. def.* **D)** *D. tri.* ***: p<0.001.

Interestingly, however, our data show that this phenomenon is only seen in *D. mel.*, suggesting a detrimental relation only between nutrition and a very high level of fecundity. As this relationship is not observed in other species, it is likely that this is not an effect of the relationship between TOR and longevity, but rather an outlier. This idea is supported by our *D. melanogaster* data on lower yeast quantities.

The interspecific differences in the longevity and fecundity data prompted us to ask if these differences were due to differential food uptake, as some fly species may eat less yeast because they prefer different kinds of food in nature. We thus carefully checked all food plates after each 24-h feeding period. On dietary plates with less than the *ad libitum* yeast quantity, the

**Table 1. Female longevity responses of four *Drosophila* species in response to four dietary treatments.**

| Species and treatment | n | mean | SEM | p-value | median | range | skew | kurtosis |
|---|---|---|---|---|---|---|---|---|
| *D. mel.* | | | | | | | | |
| no yeast | 26 | 19.12 | 1.01 | 0.0001 | 19.50 | 27 | -1.24 | 3.97 |
| Minimum | 27 | 28.44 | 2.92 | - | 28.00 | 59 | 0.61 | 0.47 |
| *ad libitum* | 30 | 23.37 | 3.12 | >0.05 | 23.00 | 66 | 0.66 | 0.19 |
| *ad libitum* with rapa | 31 | 12.13 | 1.14 | $1.1 \times 10^{-7}$ | 12.00 | 25 | 0.06 | -0.60 |
| *D. gut.* | | | | | | | | |
| no yeast | 30 | 12.40 | 1.03 | $6.8 \times 10^{-7}$ | 14.50 | 18 | -0.46 | -1.08 |
| Minimum | 30 | 30.30 | 3.76 | - | 25.00 | 74 | 0.85 | 0.03 |
| *ad libitum* | 30 | 55.60 | 5.60 | $3.5 \times 10^{-4}$ | 51.00 | 124 | 0.48 | 0.41 |
| *ad libitum* with rapa | 30 | 28.03 | 3.95 | >0.5 | 17.00 | 76 | 0.81 | -0.50 |
| *D. def.* | | | | | | | | |
| no yeast | 29 | 8.17 | 0.35 | $5.1 \times 10^{-9}$ | 8.00 | 9 | -0.20 | 1.40 |
| Minimum | 30 | 16.87 | 1.38 | - | 16.00 | 30 | 0.36 | -0.58 |
| *ad libitum* | 30 | 36.43 | 3.68 | $2.9 \times 10^{-7}$ | 34.50 | 75 | 0.08 | -0.68 |
| *ad libitum* with rapa | 30 | 27.47 | 2.47 | $3.7 \times 10^{-5}$ | 27.00 | 53 | -0.17 | -0.13 |
| *D. tri.* | | | | | | | | |
| no yeast | 26 | 13.19 | 0.62 | $8.6 \times 10^{-7}$ | 13.50 | 15 | -2.36 | 8.17 |
| Minimum | 29 | 32.48 | 4.36 | - | 25.00 | 89 | 1.25 | 0.85 |
| *ad libitum* | 29 | 60.31 | 5.33 | $1.6 \times 10^{-4}$ | 57.00 | 118 | 0.08 | -0.17 |
| *ad libitum* with rapa | 29 | 68.24 | 4.17 | $4.5 \times 10^{-6}$ | 75.00 | 71 | -0.38 | -1.09 |

Measures of female lifespan in [days] (mean and median) for *D. mel.*, *D. gut.*, *D. def.*, and *D. tri.* treated with diets ranging from no yeast, minimum (1 scoop/fly/day), to *ad libitum*, as well as an *ad libitum* yeast diet on rapamycin-treated medium (200 μM). n = number of flies going into the analysis; escaped or flies killed by accident were discarded. The mean and median values are given in [days]. p-Values were obtained by comparing test conditions to minimum feeding of that species and analyzed using pairwise comparison using Log-Rank.

food was always completely eaten, and the plates were covered in comparable amounts of feces. The *ad libitum* plates also contained equally large amounts of feces across the four species, although there was of course yeast left in all of them, as the *ad libitum* amount was meant to leave plenty of yeast left on the plates every day. Thus, the observations suggest that the observed phenotypic differences were due to post-feeding events, hinting at differences in TOR pathway regulation as this pathway is central to the regulation of energy expenditure in eukaryotic organisms [3]. The following experiments will provide deeper insight into the properties of yeast before we explore the involvement of TOR signaling.

## Examination of the effects of dead yeast on fecundity and longevity of the four *Drosophila* species

To further explore the attractiveness of Baker's yeast to our four *Drosophila* species and the eagerness of the flies to consume comparable amounts of yeast, we performed an additional experiment to test the effects of live versus dead Baker's yeast. In this experiment, we used the same Bakers' yeast but heat-treated it in order to kill the yeast while maintaining the same nutritional value. The heat treatment was performed in a similar way as done by Grangeteau et al. (2018), who showed that juvenile flies reared on heated yeast experienced a significant reduction in adult lifespan [44]. We observed that three of the four species showed no interest in the heat-treated Baker's yeast, leaving it mostly uneaten on the experimental plates: *D. mel.*, *D. gut.*, and *D. tri.* The refusal to eat dead yeast was also evident in the longevity data shown in

**Table 2. Male longevity responses of four *Drosophila* species in response to four dietary treatments.**

| Species and treatment | n | mean | SEM | p-value | median | range | skew | kurtosis |
|---|---|---|---|---|---|---|---|---|
| *D. mel.* | | | | | | | | |
| no yeast | 30 | 16.17 | 0.47 | >0.05 | 16.00 | 11 | 0.15 | 0.02 |
| Minimum | 27 | 15.59 | 1.29 | - | 18.00 | 24 | -0.94 | -0.00 |
| *ad libitum* | 19 | 15.84 | 2.38 | 0.001 | 15.00 | 31 | 0.08 | -1.35 |
| *ad libitum* with rapa | 30 | 21.10 | 2.77 | >0.05 | 16.00 | 56 | 0.94 | -0.15 |
| *D. gut.* | | | | | | | | |
| no yeast | 30 | 10.57 | 0.95 | $7.3 \times 10^{-4}$ | 12.00 | 20 | -0.14 | -0.82 |
| minimum | 30 | 22.50 | 3.50 | - | 16.50 | 70 | 1.33 | 1.10 |
| *ad libitum* | 30 | 44.77 | 4.58 | $8.5 \times 10^{-4}$ | 44.00 | 96 | 0.13 | -0.57 |
| *ad libitum* with rapa | 30 | 29.03 | 4.44 | >0.1 | 20.00 | 102 | 1.73 | 2.91 |
| *D. def.* | | | | | | | | |
| no yeast | 30 | 7.40 | 0.45 | $1.9 \times 10^{-7}$ | 7.00 | 11 | -0.45 | 0.82 |
| minimum | 30 | 18.07 | 2.30 | - | 15.50 | 53 | 1.42 | 2.25 |
| *ad libitum* | 28 | 73.64 | 4.24 | $2.3 \times 10^{-13}$ | 74.50 | 96 | -0.85 | 0.79 |
| *ad libitum* with rapa | 29 | 35.38 | 2.54 | $4.5 \times 10^{-5}$ | 36.00 | 60 | -0.58 | 0.55 |
| *D. tri.* | | | | | | | | |
| no yeast | 30 | 11.20 | 0.47 | $1.1 \times 10^{-7}$ | 11.00 | 12 | -0.29 | 0.63 |
| minimum | 31 | 28.55 | 3.97 | - | 16.00 | 64 | 1.06 | -0.53 |
| *ad libitum* | 29 | 54.76 | 4.75 | $9.6 \times 10^{-5}$ | 58.00 | 91 | -0.55 | -0.63 |
| *ad libitum* with rapa | 29 | 57.31 | 5.61 | $3.2 \times 10^{-5}$ | 63.00 | 104 | -0.48 | -0.74 |

Measures of male lifespan in [days] (mean and median) for *D. mel.*, *D. gut.*, *D. def.*, and *D. tri.* treated with diets ranging from no yeast, minimum (1 scoop/fly/day), to *ad libitum*, as well as an *ad libitum* yeast diet on rapamycin-treated medium (200 μM). n = number of flies going into the analysis; escaped or flies killed by accident were discarded. The mean and median values are given in [days]. p-Values were obtained by comparing test conditions to minimum feeding of that species and analyzed using pairwise comparison using Log-Rank.

Table 3, where the dead yeast resulted in similar lifespans as the no-yeast treatment did, while the alive-yeast treatment resulted in mean lifespans roughly twice as long as the dead and no-yeast treatments. The exception was *D. def.*, which consumed both dead and live yeast indiscriminately, which was also reflected in the longevity results, where the mean longevity on dead yeast resembled that of live yeast, both living twice as long as on no yeast.

The fecundity data on dead Baker's yeast showed across the board that dead yeast resulted in strongly reduced egg production than when the same amount of live yeast was present (Fig 3). *D. mel.* laid an average of one egg per female per day over the first 10 reproductive days on dead yeast, as compared to four eggs on live yeast, whereas the three non-model species laid no eggs at all on dead yeast. This result is likely due to the rejection of the dead yeast, perhaps because it does not produce appetite-inducing fermentation products. While we did not further investigate the fertility of these eggs, Grangeteau et al. (2018) showed that out of 50 eggs reared on the heated yeast diet, slightly less than two-thirds made it into adulthood [44]. What is less intuitive in our experiment was the fact that *D. def.* did not produce eggs even though it ate the dead yeast. As this species feeds on dead leaf matter in nature [31, 32], it may not be bothered by the lack of fermentation products and eat what is offered; however, the lack of egg production leads us to speculate that some of the key nutrients within the yeast may have been damaged during the baking process. Potentially, the consumption of dead yeast is sufficient for sustaining life; however, not favorable enough to activate the TOR system for the further production of offspring.

**Table 3. Longevity of four *Drosophila* species in response to dead Baker's yeast.**

| | Female | | | Male | | |
|---|---|---|---|---|---|---|
| | **mean** | **SEM** | **p-value** | **mean** | **SEM** | **p-value** |
| *D. mel.* | | | | | | |
| no yeast | 19.115 | 1.014 | - | 16.167 | 0.465 | - |
| dead yeast | 20.517 | 1.832 | 0.021 | 18.107 | 1.385 | $3.1 \times 10^{-4}$ |
| minimal alive yeast | 28.444 | 2.921 | $2.2 \times 10^{-4}$ | 15.5923 | 1.293 | 0.090 |
| *D. gut.* | | | | | | |
| no yeast | 12.4 | 1.029 | - | 10.567 | 0.948 | - |
| dead yeast | 15.733 | 1.638 | 0.041 | 13.733 | 1.334 | 0.031 |
| minimal alive yeast | 30.3 | 3.761 | $6.8 \times 10^{-7}$ | 22.5 | 3.501 | 0.001 |
| *D. def.* | | | | | | |
| no yeast | 8.172 | 0.355 | - | 7.4 | 0.449 | - |
| dead yeast | 14.690 | 1.135 | $1.1 \times 10^{-7}$ | 16.133 | 1.626 | $1.1 \times 10^{-7}$ |
| minimal alive yeast | 16.867 | 1.378 | $7.7 \times 10^{-9}$ | 18.067 | 2.297 | $2.4 \times 10^{-7}$ |
| *D. tri.* | | | | | | |
| no yeast | 13.192 | 0.620 | - | 11.2 | 0.468 | - |
| dead yeast | 14.333 | 0.974 | 0.55 | 11.148 | 0.695 | 0.86 |
| minimal alive yeast | 32.483 | 4.360 | $1.3 \times 10^{-6}$ | 28.548 | 3.972 | $1.6 \times 10^{-7}$ |

Measures of female and male lifespan of *D. mel.*, *D. gut.*, *D. def.*, and *D. tri.* treated with diets including no yeast, minimum dead yeast (1 scoop/fly/day), and minimum alive yeast (1 scoop/fly/day). All unequal variance t-tests were performed between no yeast and the other two treatments per species. p-Values were obtained by comparing test conditions to no yeast diets of that species and analyzed using pairwise comparison using Log-Rank.

## Effects of TOR pathway inhibition by rapamycin in four *Drosophila* species

After having verified that live instant Baker's yeast is readily consumed by all four *Drosophila* species, we next asked the question if differential TOR pathway signaling is responsible for the observed species-specific fecundity level and longevity phenotypes among the four *Drosophila* species. We thus blocked the TOR pathway by adding rapamycin to the food, while exposing the flies to different quantities of live Baker's yeast [23]. Our hypothesis was that if the TOR pathway regulates longevity and/or fecundity, then the inhibitor rapamycin should reverse the effects that we observed when overfeeding the flies on an *ad libitum* diet and send the cells into maintenance and cleanup mode, resulting in a decrease in fecundity and a restoration of longevity lost due to overfeeding, as we observed in *D. mel.* females [21, 22, 24]. Female flies of all four species showed indeed an at least 4-fold reduction in their fecundity rates within the first 10 reproductive days, when they were exposed to rapamycin on an *ad libitum* diet (Fig 4). The fecundity of *D. mel.* dropped from almost 40 eggs per female per day to around 10; *D. gut.* dropped from 15 to 4; *D. tri.* dropped from 7 to 2; while *D. def.* stopped producing eggs altogether. The observed reduction in fecundity was clearly not due to a lack of feeding activity on the rapamycin plates, as the amount of feces was comparable to the plates without rapamycin. However, the feces on rapamycin plates often looked watery with visible granules, which could be a side effect of rapamycin. It was previously reported that in mice, rapamycin led to smaller, pellet-like feces or diarrhea, possibly due to the restructuring of the microbiome [45, 46]. In order to verify our conclusion that the reduction in fecundity was due to the addition of rapamycin and not due to denial of food, we repeated this experiment with *D. mel.* on a medium yeast diet (2 scoops/fly/day), with and without rapamycin. This time, we confirmed that rapamycin did not lead to the rejection of the yeast (the yeast was eaten), while the fecundity was again decreased, this time by about half (Fig 5). In previous studies, the effects of rapamycin

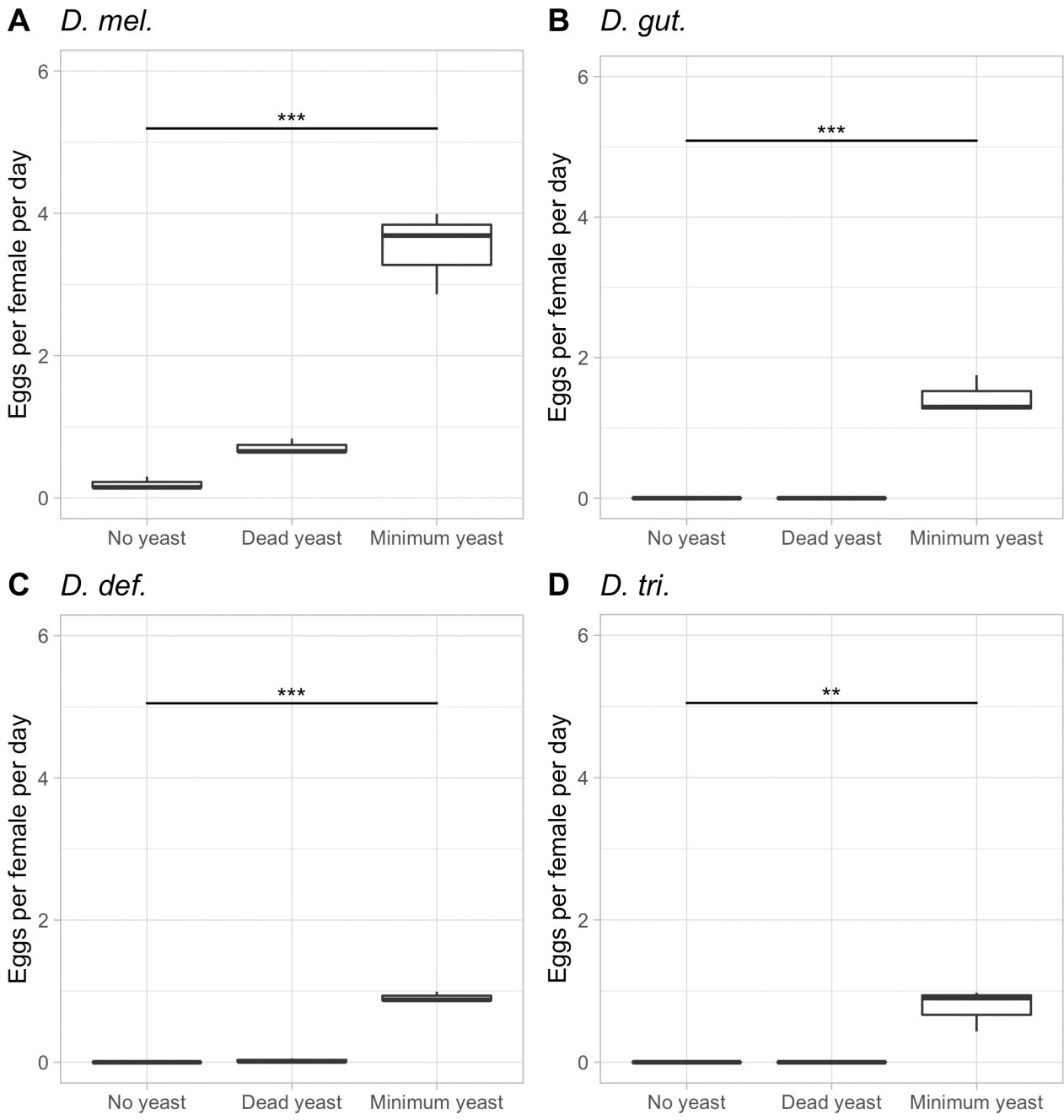

**Fig 3. Egg-lay responses of the four *Drosophila* species in response to dead yeast.** The average egg lay response over the first 10 reproductive days of *D. mel.* treated with diets including no yeast, minimum dead yeast (1 scoop/fly/day), and minimum alive yeast (1 scoop/fly/day) are shown. p< 0.001: '***' p<0.01: '**'.

on whole bodies as well as cellular intake of nutrients had been investigated. Similar to our study, the addition of rapamycin did not alter food intake but led to a change in the response to nutrients, resulting in a reduction of the adipose tissue and in decreased body weight in mice [47, 48].

To shed light on the anatomical causes for the variation in fecundity in *D. mel.*, we tested if different yeast quantities and the presence of rapamycin affected ovary development. We

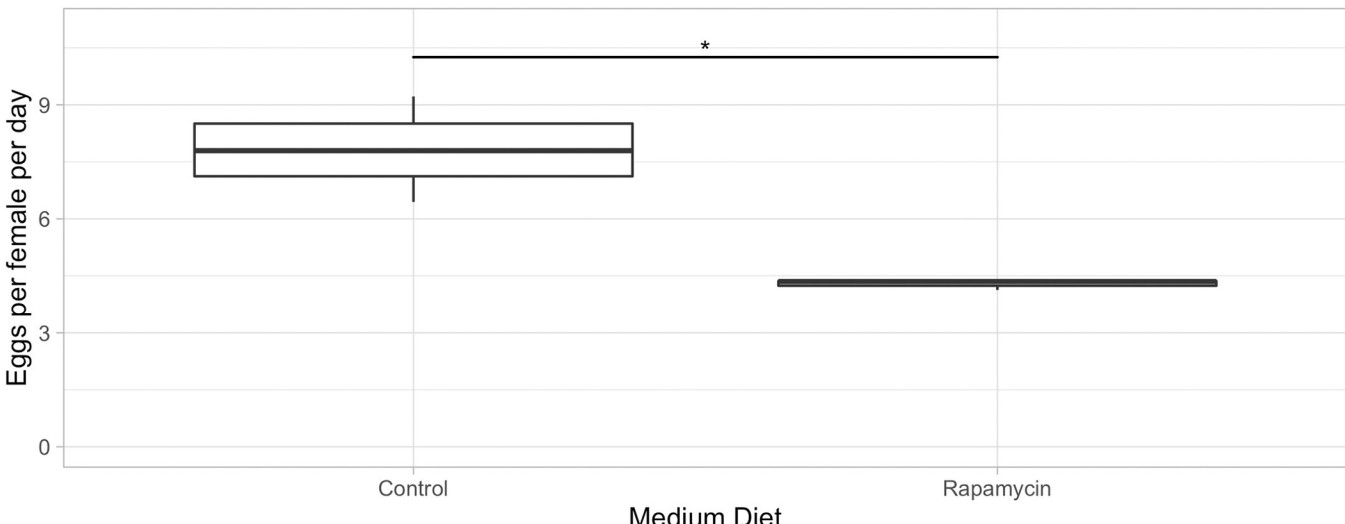

**Fig 4. Female fecundity in response to rapamycin in four _Drosophila_ species on _ad libitum_ diets.** The average egg-lay response across the first ten reproductive days of _D. mel._, _D. gut._, _D. def._, and _D. tri._ on an _ad libitum_ diet, with or without the addition of rapamycin, are shown. **A)** _D. mel._ **B)** _D. gut._ **C)** _D. def._ **D)** _D. tri._ p<0.001: '***', p< 0.01 = '**'.

**Fig 5. Female fecundity in response to rapamycin in _D. mel._ on a medium diet.** The average egg-lay response over the first 10 reproductive days of _D. mel._ treated with a medium yeast diet (2 scoops/fly/day) with or without rapamycin. p<0.05: '*'.

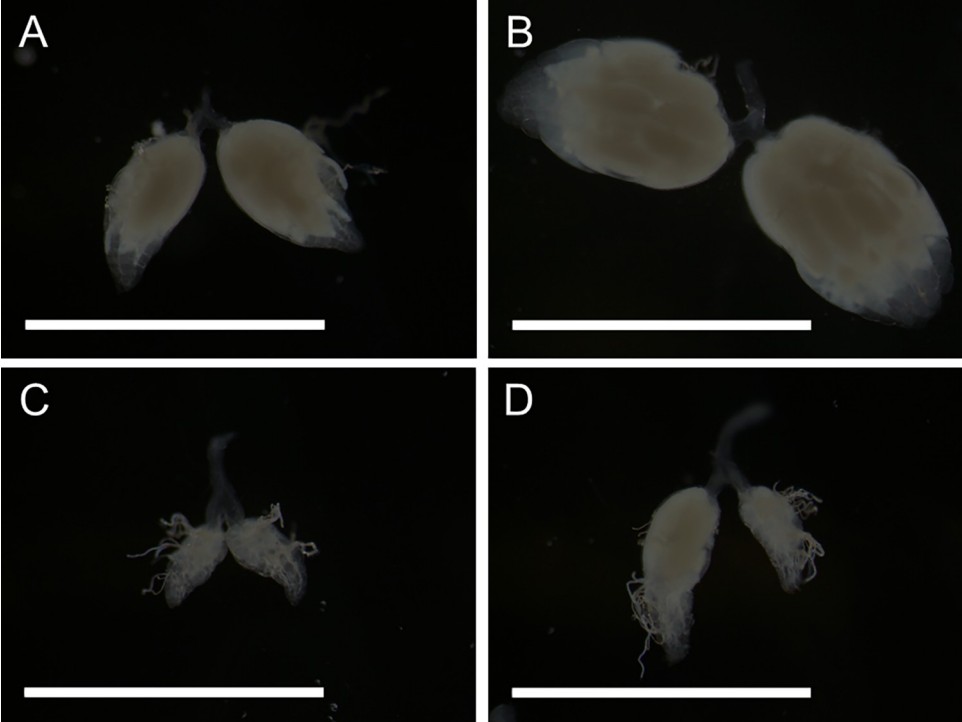

**Fig 6. Ovary dissections of *D. mel*. on four different diets.** All images were taken at 25X magnification. The white bars measure 0.5 mm. **A)** *D. mel*. ovaries on a medium diet (2 scoops/fly/day). **B)** ovaries on an *ad libitum* yeast diet. **C)** ovaries on a medium diet with rapamycin. **D)** ovaries on an *ad libitum* yeast diet with rapamycin.

treated *D. mel*. flies with a medium yeast diet (2 scoops/fly/day) as well as an *ad libitum* diet, each with and without rapamycin. We then dissected the ovaries of five females per treatment and imaged one representative pair of ovaries for each treatment. All flies were five days old at the time of dissection. As shown in Fig 6, *D. mel*. females on a medium diet showed medium-sized ovaries (Fig 6A), whereas the overfed flies' ovaries increased about three-fold (Fig 6B), explaining the higher egg-lay capacity on an *ad-libitum* diet, as compared to the more restricted diet. When rapamycin was added, the ovaries on the medium diet appeared highly underdeveloped (Fig 6C), which agrees with the low egg-lay rate (Fig 5), while on an *ad libitum* diet with rapamycin, the ovaries were somewhat bigger but showed some asymmetry caused by one side having a nearly mature egg in the ovary, while the other side does not. These data agree with those obtained with previously published hypomorphic *Tor* mutant data, resulting in small and distorted ovaries [19]. We conclude that both nutrients and TOR signaling positively impact ovary development.

From the data seen thus far for *D. mel*. (Figs 4A, 5 and 6), the species that will be our focus for a moment, it appears that the most drastic fecundity reduction through TOR signal inhibition is evident on the *ad libitum* yeast diet, at which the egg productivity is at its maximum without TOR inhibition. Our next question was if rapamycin can reverse the longevity decline that *D. mel*. females experienced on the *ad libitum* diet. We thus analyzed the longevity data from the same experiment, for which we presented the fecundity data in Fig 4. To our surprise, rapamycin did not increase but instead reduced female longevity by approximately half in *D. mel*. females fed with the *ad libitum* diet supplemented with rapamycin (Table 1, S2 Fig), while the longevity of the *D. mel*. males on the same diet only slightly decreased (Table 2, S2 Fig). As we know from the literature, a combination of rapamycin at 200 μM and moderate diet of 100

g/L of yeast in the agar has been shown to extend the lifespan of *D. mel.* [6, 23]. Thus, the lack of the expected lifespan-extending effect of rapamycin of *D. mel.* females in our experiment may be due to the *ad libitum* diet itself because Bjedov et al. (2010) showed that the longevity-extending effect of rapamycin diminished as the nutrients fed to the flies increased under a constant rapamycin concentration [23]. Thus, our *ad libitum*-fed *D. mel.* females may have had too many nutrients to observe a longevity-increasing effect of rapamycin, even when the egg-lay productivity was strongly reduced, and thereby the reproductive stress. The open question that remains is why rapamycin further decreased the lifespan of *D. mel.* females below the levels seen on the *ad libitum* diet alone (Table 1). Although no mechanism has been proposed, Harrison et al. (2010) found that when the rapamycin concentration is increased between 0–1000 μM on a 1:1 yeast: sugar diet, the longevity of female flies decreased [36]. Furthermore, when Villa-Cuesta et al. (2014) fed flies on a 12% yeast diet with rapamycin ranging from 0–400 μM, the flies showed no extended lifespan due to rapamycin. However, on a 2% yeast diet, rapamycin concentrations of 50–200 μM rapamycin did extend the longevity, suggesting the possible need for a moderated diet in order to achieve a life-prolonging effect through the use of rapamycin [25]. It is therefore possible that our choice of a 200 μM final rapamycin concentration was high enough to observe phenotypic effects due to TOR pathway blockage but too high to allow *D. mel.* females to restore their longevity on an *ad libitum* diet. We followed up with a small experiment to test if lower yeast quantities show the desired lifespan increase due to rapamycin on a medium yeast diet (2 scoops/fly/day) and 200 μM of rapamycin, but we were still unable to increase female longevity (Table 4). Instead the longevity of both males and females slightly declined with the rapamycin treatment; however, not significantly so (Table 4, S4 Fig, S1 File). We decided, however, not to decrease our rapamycin concentration across our experiments further because 1) on lower rapamycin concentrations, we did not detect significant phenotypic effects of the drug in our pilot experiments and 2) we needed to make sure that all four of our chosen species show phenotypic effects of the rapamycin treatment, which was the case with the chosen 200 μM final rapamycin concentration. We will now discuss the data of the effects of rapamycin on the three non-model *Drosophila* species previously included in Tables 1 and 2.

Rapamycin had the following effects on the longevity and fecundity of the three non-model species on an *ad libitum* diet: the longevity of *D. gut.* significantly decreased in both sexes to levels comparable to receiving the minimum amount of yeast (without rapamycin) (Tables 1 and 2), while female fecundity was reduced by about 5-fold (Fig 4B). *D. def.* behaved in a similar fashion: the longevity significantly declined in both sexes to higher levels than those achieved by receiving the minimum amount of yeast without rapamycin (Tables 1 and 2). The egg production stopped entirely in *D. def.* in response to rapamycin (Fig 4B). *D. tri.* is the only

**Table 4.  Longevity respsonses of *D. mel.* to medium diet with and without rapamycin.**

| Males | n | mean | SEM | p-value | median | range | skew | kurtosis |
|---|---|---|---|---|---|---|---|---|
| Medium diet | 28 | 18.39 | 1.56 | - | 18 | 32 | -0.01 | -0.68 |
| Medium diet with rapa | 29 | 15.62 | 1.58 | 0.038 | 14 | 34 | 1.04 | 0.86 |
| Females | | | | | | | | |
| Medium diet | 28 | 30.14 | 3.56 | - | 25 | 74 | 0.84 | 0.10 |
| Medium diet with rapa | 29 | 21.21 | 2.58 | 0.35 | 19 | 61 | 1.14 | 1.88 |

Measures of male and female lifespan in [days] (mean and median) for *D. mel.* on medium (2 scoops/fly/per day) diets with and without rapamycin (200 μM).

n = number of flies going into the analysis; escaped or flies killed by accident were discarded. The mean and median values are given in [days]. p-Values were obtained by comparing the diet with and without rapamycin and analyzed using pairwise comparison using Log-Rank.

species of the four tested, in which we saw the anticipated lifespan-enhancing effect of rapamycin [23]. Both sexes increased their lifespan by roughly 10% (Tables 1 and 2), while female fecundity decreased by approximately 4-fold (Fig 4B). Although we do not have any molecular data on TOR signaling, we interpret the differences in longevity and fecundity seen among the four *Drosophila* species in response to TOR inhibition as an indication that TOR signaling may have changed over evolutionary time among *Drosophila* species. While there are similar trends across all four species, their independent evolutionary paths may have driven changes in their responses to nutrients, leading to changes in fecundity levels and accompanying lifespans.

## Implications

Our results suggest the existence of species-specific differences in the response to varying amounts of yeast and dietary rapamycin in four species of the genus *Drosophila*. While these species are all relatively closely related, they seem to exhibit different levels of fecundity and longevity when exposed to similar dietary environments. Here we hypothesize that these differences might in part be due to evolutionary changes in the regulation of the TOR pathway. Despite the TOR pathway being deeply conserved among virtually all eukaryotic organisms, our results resulting from activating TOR through the diet and inhibiting it with rapamycin seem to point to slight differences in TOR regulation. In response to TOR pathway manipulation through the variations in diet and inhibition by rapamycin, some thriving species saw large differences in lifespan and fecundity, while others remained consistant. A thorough understanding of TOR signaling, particularly its dysregulation, is important to guide the treatment of many human diseases, such as obesity, cancer, and metabolic syndrome [49, 50]. Our research emphasizes the importance of incorporating non-model organisms into modern research, as they can provide us with natural genetic variance that the standard model organisms are clearly lacking.

## Limitations

In order to fully understand TOR pathway regulation and evolution, it would be necessary to look at the regulatory control of this pathway at the genomic, transcriptomic, and proteomic levels, which was outside the scope of the current study. We instead decided to primarily investigate the effects of activating TOR through the diet and to observe wholistic outcomes, in particular longevity and fecundity, which are easy to measure. Our observations may inspire further genetic and metabolomic analyses to find the genetic loci at which the evolutionary changes in the regulation of TOR signaling took place.

Furthermore, the knowledge about most non-model *Drosophila* species is still very limited, and many species are difficult to rear, making it harder to perform experiments with them. We chose *D. mel.*, *D. gut.*, *D. def.*, and *D. tri.* for this study because they have been of previous interest to our lab and can be successfully reared on the same fly food as well as on yeast alone. Because we have maintained these flies for several years in our lab, they have adapted to laboratory food conditions, which may have influenced the effects that we detected in our experiments. A way to circumvent this problem would be to collect wild flies in nature and to immediately sequence their DNA and RNA to draw conclusions about TOR pathway activity, although it would not allow to test for fecundity and longevity, nor would it be possible to know what the flies were feeding on prior to their capture.

In addition to the flies' original wild locations, there may also be several other factors that influence the connection between nutritional intake, activation of metabolism, and their longevity and fertility. One of the primary differences is likely the microbiome of the flies. The

microbiome has been shown to affect lifespan and fecundity [51]. In the lab, the microbiome is mostly obtained from the parents and the lab environment, with bacteria belonging to the families Acetobacteracceae and Lactobacillaceae being the most prevalent components [52–54]. It is likely that the microbiomes of our older lab strains *D. mel*, *D. gut*., and *D. def*. are comparable, while that of the more recently wild-caught *D. tri*. might more closely resemble the natural microbiome typical for the region in which it was caught. The digestive capacity of the flies might inherently be distinct as well. The flies are all different in size and activity, perhaps requiring incomparable amounts of food. These differences could not be controlled for in this experiment, but may nevertheless impact longevity and fertility.

Another confounding factor could be the mating frequency. In *Drosophila suzukii* (the Spotted Wing Drosophila or SWD), it has been shown that exposure to higher yeast leads to more mated female flies, but it did not lead to an increase in oviposition [55]. While a link between mating frequency and a decrease in lifespan has been proposed [56], it is still heavily debated in the *Drosophila* literature [57]. Further experiments focused either on female or male longevity should control for mating activity on different yeast quantities.

Even though we followed the entire adult life spans of our experimental flies, it would have been ideal to be able to determine not only fecundity (the number of eggs laid), but also fertility (viable offspring, e.g., number of hatching larvae or of eclosed flies in the next generation). Perhaps fertility data would give more detail about the condition of the eggs and the females' ability to create successful offspring. We note that in our pilot studies, fertility rates (as measured in hatched larvae from the eggs) were often around 70% for all species across all diets.

### Future research

Based on the current capability to compare species on the genomic, transcriptomic, and proteomic levels, the extension of this research is to identify differential TOR pathway signaling events caused by evolutionary changes among model and non-model *Drosophila* species with different lifestyles and dietary needs. This research would be geared towards the greater goal of investigating the functional evolution of TOR and may result in the establishment of new model species for common metabolic diseases. As TOR signaling is also known to cross-talk with the *Drosophila* microbiome, having implications on larval growth rates in response to nutrient availability [58], the microbiome deserves special attention when trying to elucidate TOR signaling differences among *Drosophila* species.

### Conclusions

This study provided clues about the existence of natural variations in TOR signaling, which may have allowed model and non-model *Drosophila* species to adapt to different food sources and ecological niches. The differences among the four *Drosophila* species studied here expressed themselves in differential responses measured in fecundity and longevity when treated on identical diets. TOR pathway inhibition reversed the phenotypes observed through TOR activation through an increased diet, which was most evident in the fecundity data. Taken together, our experiments exposed the previously hidden potential of non-model species to study TOR pathway signaling diversity and evolution, which may have a future impact on medical research to solve problems related to overeating, such as obesity and metabolic syndrome.

### Supporting information

**S1 Fig. Survival curves of *D. mel*. *D. mel*. females and males on four diets: No yeast, ¼ scoop of yeast (62.5 nL) per fly per day, ½ scoop per fly per day, 1 scoop per fly per day,**

**and *ad libitum* yeast.**
(TIF)

**S2 Fig. Survival curves of four *Drosophila* species in response to four dietary treatments.** Survival curves for *D. mel.*, *D. gut.*, *D. def.*, and *D. tri.* treated with diets ranging from no yeast (purple), minimum (1 scoop/fly/day, blue), to *ad libitum* (red), as well as an *ad libitum* yeast diet on rapamycin-treated medium (200 μM, green). Escaped or flies killed by accident were discarded.
(TIF)

**S3 Fig. Survival curves of four *Drosophila* species in response to dead Baker's yeast.** Female and male lifespan of *D. mel.*, *D. gut.*, *D. def.*, and *D. tri.* treated with diets including no yeast (blue), minimum dead yeast (1 scoop/fly/day, red), and minimum alive yeast (1 scoop/fly/day, green).
(TIF)

**S4 Fig. Survival curves of *D. mel.* on a medium diet with and without rapamycin.** Survival curves for male and female lifespan in [days] for *D. mel.* on medium (2 scoops/fly/per day) diets with (blue) and without (red) rapamycin (200 μM). n = number of flies going into the analysis; escaped or flies killed by accident were discarded.
(TIF)

**S1 File. Suvival curve Log-Rank results.** p-values for all compared survival curves regarding longevity are shown.
(XLSX)

## Acknowledgments

We thank Lucinda Hall, Madeline Kmieciak, Karmyn Polakowki, and Morgan Smith for their continuous support in the ongoing experiments.

## Author Contributions

**Conceptualization:** Tessa E. Steenwinkel, Thomas Werner.

**Data curation:** Tessa E. Steenwinkel, Kailee K. Hamre, Thomas Werner.

**Formal analysis:** Tessa E. Steenwinkel, Kailee K. Hamre.

**Funding acquisition:** Tessa E. Steenwinkel, Thomas Werner.

**Investigation:** Tessa E. Steenwinkel, Kailee K. Hamre, Thomas Werner.

**Methodology:** Tessa E. Steenwinkel, Thomas Werner.

**Project administration:** Tessa E. Steenwinkel, Thomas Werner.

**Resources:** Tessa E. Steenwinkel, Thomas Werner.

**Software:** Tessa E. Steenwinkel.

**Supervision:** Thomas Werner.

**Validation:** Tessa E. Steenwinkel, Thomas Werner.

**Visualization:** Tessa E. Steenwinkel, Thomas Werner.

**Writing – original draft:** Tessa E. Steenwinkel, Kailee K. Hamre, Thomas Werner.

**Writing – review & editing:** Tessa E. Steenwinkel, Kailee K. Hamre, Thomas Werner.

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
