## [Decision Letter · Decision Letter 0]

8 Jul 2022

PONE-D-22-16667The use of non-model Drosophila species to study natural variation in TOR pathway signalingPLOS ONE

Dear Dr. Werner,

Thank you for submitting your manuscript to PLOS ONE. After careful consideration, we feel that it has merit but does not fully meet PLOS ONE’s publication criteria as it currently stands. Therefore, we invite you to submit a revised version of the manuscript that addresses the points raised during the review process.

We look forward to receiving your revised manuscript.

Kind regards,

Kyung-Jin Min

Academic Editor

PLOS ONE

Journal Requirements:

"We thank Lucinda Hall, Madeleine Kmieciak, Karmyn Polakowski, and Morgan Smith for their continuous support in the ongoing experiments. This research was funded by NSF grant #DOB/DEB1737877 to T.W. as well as the Barry Goldwater Fellowship, a Michigan Tech SURF Award, and a Michigan Tech Songer Research Award to T.E.S. "

"This research was funded by NSF grant #DOB/DEB1737877 to T.W. as well as the Barry Goldwater Fellowship, a Michigan Tech SURF Award, and a Michigan Tech Songer Research Award."

Reviewers' comments:

Reviewer's Responses to Questions

**Comments to the Author**

1. Is the manuscript technically sound, and do the data support the conclusions?

Reviewer #1: Partly

Reviewer #2: Yes

2. Has the statistical analysis been performed appropriately and rigorously? 

Reviewer #1: No

Reviewer #2: No

3. Have the authors made all data underlying the findings in their manuscript fully available?

Reviewer #1: Yes

Reviewer #2: Yes

4. Is the manuscript presented in an intelligible fashion and written in standard English?

Reviewer #1: Yes

Reviewer #2: Yes

5. Review Comments to the Author

Reviewer #1: This manuscript studies the fecundity and longevity of four different species of Drosophila. The main data shown by the authors is that as yeast concentration increases so does fecundity. The authors use an inhibitor of TOR pathway known to decrease fecundity to show that fecundity is inhibited when TOR pathway is inhibited. They also showed mean and median longevity studies. However, little can be concluded from the longevity studies since no statistical power or survival analysis are provided. The data does not seem to support the claim about how non-model Drosophila species showed natural variation in TOR signaling.

Reviewer #2: In the manuscript, the authors examined the interspecific differences in nutrient signaling among four difference Drosophila species. Results showed that there may be natural variation in the TOR signaling based on Fecundity and longevity data. Results are interesting and there seems to be no major errors in experimental design and/or analysis. I have just minor suggesitons.

1. Only 30 flies were tested in each group which will weaken the statistical analysis. Any reason not to increase the replicate number?

2. More rigorous statistical analysis for longevity assay is necessary - there is no p values

3. Even though the authors compared the feeding rate among four Drosophila species, the differences in life-history data may be due to other factors like differences in microbiome and/or differences in digestive capacity. It should be discussed somewhere.

4. It seems that there were 10 males and 10 females in a single vial. Mating frequency is different between low and high yeast diets which will also affect longevity. The confounding effect of mating should be discussed also.

6. PLOS authors have the option to publish the peer review history of their article (what does this mean?). If published, this will include your full peer review and any attached files.

Reviewer #1: No

Reviewer #2: **Yes: **Kyung-Jin Min

---

## [Author Response · Author response to Decision Letter 0]

19 Aug 2022

Response to reviewers

PONE-D-22-16667

The use of non-model Drosophila species to study natural variation in TOR pathway signaling

Reviewer #1: 

This manuscript studies the fecundity and longevity of four different species of Drosophila. The main data shown by the authors is that as yeast concentration increases so does fecundity. The authors use an inhibitor of TOR pathway known to decrease fecundity to show that fecundity is inhibited when TOR pathway is inhibited. They also showed mean and median longevity studies. 

Comment: However, little can be concluded from the longevity studies since no statistical power or survival analysis are provided. The data does not seem to support the claim about how non-model Drosophila species showed natural variation in TOR signaling.

Response: To address the lack of statistical analysis in the form of survival analysis and accompanying p-values, supplemental files were added showing all survival curves for the in-text tables. Additional p-values were included in in-text charts where appropriate and attached as a supplemental file. We also toned down the observation about the TOR regulation.

Reviewer #2: 

In the manuscript, the authors examined the interspecific differences in nutrient signaling among four difference Drosophila species. Results showed that there may be natural variation in the TOR signaling based on Fecundity and longevity data. Results are interesting and there seems to be no major errors in experimental design and/or analysis. I have just minor suggestions.

Comment 1: Only 30 flies were tested in each group which will weaken the statistical analysis. Any reason not to increase the replicate number?

Response 1: The flies were split up into three replicates, each containing 10 males and 10 females. The flies were housed in small Petri dishes and anesthetized by hand to sort out any deaths and move them to the next day’s Petri dish. After moving the flies, the eggs were counted by hand. Due to the desire to have telling results while having manageable egg numbers, total female flies were limited to 30. Future projects could take advantage of automated counting. 

Comment 2: More rigorous statistical analysis for longevity assay is necessary - there is no p values.

Response 2: To address the lack of statistical analysis in the form of survival analysis and accompanying p-values, five supplemental files (S1 – S4_Figs and S5_File) were added showing all survival curves for the in-text tables. Additional p-values were included in in-text charts where appropriate and attached as a supplemental file

Comment 3: Even though the authors compared the feeding rate among four Drosophila species, the differences in life-history data may be due to other factors like differences in microbiome and/or differences in digestive capacity. It should be discussed somewhere.

Response 3: To address the comment about the possible influence of the microbiota and/or digestive capacity, a paragraph was added to the limitations. In our long-term lab reared species, we believe that the microbiomes are similar, while more recent wild-caught flies likely resemble the natural state of the original region. 

Comment 4: It seems that there were 10 males and 10 females in a single vial. Mating frequency is different between low and high yeast diets which will also affect longevity. The confounding effect of mating should be discussed also.

Response 4: We included a new section addressing these issues in the discussion as well as a paragraph in the limitations section. The discussion of the effect of increased mating on longevity is still ongoing and not entirely understood. It is, however, a factor to consider for further experimentation.

---

## [Decision Letter · Decision Letter 1]

6 Sep 2022

The use of non-model Drosophila species to study natural variation in TOR pathway signaling

PONE-D-22-16667R1

Dear Dr. Werner,

We’re pleased to inform you that your manuscript has been judged scientifically suitable for publication and will be formally accepted for publication once it meets all outstanding technical requirements.

Kind regards,

Kyung-Jin Min

Academic Editor

PLOS ONE

Additional Editor Comments (optional):

Reviewers' comments:

Reviewer's Responses to Questions

**Comments to the Author**

1. If the authors have adequately addressed your comments raised in a previous round of review and you feel that this manuscript is now acceptable for publication, you may indicate that here to bypass the “Comments to the Author” section, enter your conflict of interest statement in the “Confidential to Editor” section, and submit your "Accept" recommendation.

Reviewer #2: All comments have been addressed

2. Is the manuscript technically sound, and do the data support the conclusions?

Reviewer #2: Yes

3. Has the statistical analysis been performed appropriately and rigorously? 

Reviewer #2: Yes

4. Have the authors made all data underlying the findings in their manuscript fully available?

Reviewer #2: Yes

5. Is the manuscript presented in an intelligible fashion and written in standard English?

Reviewer #2: Yes

6. Review Comments to the Author

Reviewer #2: All concerns were addressed and this manuscript is now qualified for the publication in PLoS One Journal

7. PLOS authors have the option to publish the peer review history of their article (what does this mean?). If published, this will include your full peer review and any attached files.

Reviewer #2: **Yes: **Kyung-Jin Min

---

## [Editor Report · Acceptance letter]

13 Sep 2022

PONE-D-22-16667R1 

The use of non-model *Drosophila* species to study natural variation in TOR pathway signaling 

Dear Dr. Werner:

I'm pleased to inform you that your manuscript has been deemed suitable for publication in PLOS ONE. Congratulations! Your manuscript is now with our production department. 

Kind regards, 

on behalf of

Dr Kyung-Jin Min 

Academic Editor

PLOS ONE